# Role of D-Mannose in the Prevention of Recurrent Uncomplicated Cystitis: State of the Art and Future Perspectives

**DOI:** 10.3390/antibiotics10040373

**Published:** 2021-04-01

**Authors:** Cosimo De Nunzio, Riccardo Bartoletti, Andrea Tubaro, Alchiede Simonato, Vincenzo Ficarra

**Affiliations:** 1Department of Urology, Sant’Andrea Hospital, Sapienza University of Rome, 00141 Rome, Italy; andrea.tubaro@uniroma1.it; 2Department of Translational Research and New Technologies, Urology Unit, University of Pisa, 56010 Pisa, Italy; riccardo.bartoletti@unipi.it; 3Department of Surgical, Oncological and Oral Sciences, Urology Section, University of Palermo, 90121 Palermo, Italy; alchiede.simonato@unipa.it; 4Department of Human and Pediatric Pathology “Gaetano Barresi”, Urology Section, University of Messina, 98168 Messina, Italy; vincenzo.ficarra@unime.it

**Keywords:** female, urinary tract infections, UTI, D-mannose, cystitis, prevention, treatment

## Abstract

Background: Urinary tract infections (UTI) are highly frequent in women, with a significant impact on healthcare resources. Although antibiotics still represent the standard treatment to manage recurrent UTI (rUTI), D-mannose, an inert monosaccharide that is metabolized and excreted in urine and acts by inhibiting bacterial adhesion to the urothelium, represents a promising nonantibiotic prevention strategy. The aim of this narrative review is to critically analyze clinical studies reporting data concerning the efficacy and safety of D-mannose in the management of rUTIs. Methods: A non-systematic literature search, using the Pubmed, EMBASE, Scopus, Web of science, Cochrane Central Register of Controlled Trials, and Cochrane Central Database of Systematic Reviews databases, was performed for relevant articles published between January 2010 and January 2021. The following Medical Subjects Heading were used: “female/woman”, “urinary tract infection”, and “D-mannose”. Only clinical studies, systematic reviews, and meta-analyses reporting efficacy or safety data on D-mannose versus placebo or other competitors were selected for the present review. Evidence was limited to human data. The selected studies were organized in two categories according to the presence or not of a competitor to D-mannose. Results: After exclusion of non-pertinent studies/articles, 13 studies were analyzed. In detail, six were randomized controlled trials (RCTs), one a randomized cross-over trial, five prospective cohort studies, and one a retrospective analysis. Seven studies compared D-mannose to placebo or others drugs/dietary supplements. Six studies evaluated the efficacy of D-mannose comparing follow-up data with the baseline. D-mannose is well tolerated, with few reported adverse events (diarrhea was reported in about 8% of patients receiving 2 g of D-mannose for at least 6 months). Most of the studies also showed D-mannose can play a role in the prevention or rUTI or urodynamics-associated UTI and can overlap antibiotic treatments in some cases. The possibility to combine D-mannose with polyphenols or *Lactobacillus* seems another important option for UTI prophylaxis. However, the quality of the collected studies was very low, generating, consequently, a weak grade of recommendations as suggested by international guidelines. Data on D-mannose dose, frequency, and duration of treatment are still lacking. Conclusion: D-mannose alone or in combination with several dietary supplements or *Lactobacillus* has a potential role in the non antimicrobial prophylaxis or recurrent UTI in women. Despite its frequent prescription in real-life practice, we believe that further well-designed studies are urgently needed to definitively support the role of D-mannose in the management of recurrent UTIs in women.

## 1. Introduction

Uncomplicated urinary tract infections (UTIs) are very common diseases affecting both sexes, with a higher prevalence in women, regardless of the age [1]. In USA, about 11% of women report at least one episode of UTI per year, and up to half of them will experience an additional UTI within the first year after the initial infection [1,2]. So far, recurrent UTI (rUTI), generally defined by at least three episodes/year or two in the last six months, have a significant impact on patients’quality of life and healthcare system resources [1]. *Escherichia coli* followed by *Proteus*, *Klebsiellas*, *Eneterococci*, *Streptococce*, and *Pseudomonas aeruginosa* are the pathogens most frequently responsible for UTIs. According to international guidelines, fosfomycin trometamol, pivmecillinam, and nitrofurantoin represent the first-line therapy of uncomplicated cystitis [3]. Conversely, guidelines strongly recommend to not use aminopenicillins or fluoroquinolones, due to the high level of antibiotic resistance currently detected in Europe [3]. At the end of a first-line antimicrobial treatment, women can show persistent symptoms or symptoms recurrence within two weeks. Although antibiotics still represent the cornerstone of UTIs management, costs, adverse events and, particularly, the increasing rise of antimicrobial resistance highlight the need for alternative prophylactic and therapeutic approaches [4]. So far, in patients with rUTIs, behavioural modifications and non-antimicrobial measures should be strongly considered before prescribing antimicrobial prophylaxis to limit its ecologic impact and risk of complications [5,6,7,8]. Hormonal replacement, immunoactive prophylaxis, probiotics, cranberry, or D-mannose and endovescical instillation were considered as common non-antimicrobial measures to prevent rUTIs. 

D-mannose represents a promising nonantibiotic prevention strategy. It is an inert monosaccharide that is metabolized and excreted in urine and acts by inhibiting bacterial adhesion to the urothelium. A simple sugar, D-mannose has an important role in human metabolism through glycosylation of proteins [9]. In details, D-mannose binds and blocks FimH adhesins located on the tip of type 1 bacterial fimbriae, playing a competitive inhibitor role against bacterial adhesion to the receptors of urothelial cells. Type 1 pili have been documented on *E. coli*, other members of the Enterobacteriaceae family including *Klebsiella pneumoniae*, *Shigella flexneri*, *Salmonella typhimurium*, *Serratia marcescens*, and *Enterobacter cloacae* [10,11]. Therefore, D-mannose can prevent the adhesion of several uropathogens involved in the UTI to the urothelium. 

For the purpose of the present scoping review of the literature, we critically analyzed clinical studies reporting data concerning the efficacy and safety of D-mannose in the management of rUTIs. The aim of the present study is to give a practical guide to urologists for the prescription of D-mannose in their clinical practice and to identify potential research areas to improve evidence in the literature. Moreover, the objective of our scoping review is to identify the types of available evidence on D-mannose activity and the key characteristics of the evidence and to analyze knowledge gaps on the subject in order to provide an overview of the evidence. Furthermore, our review could be considered a precursor of a future systematic review. 

## 2. Evidence Acquisition

A non-systematic literature search, using Pubmed, EMBASE, Scopus, Web of science, Cochrane Central Register of Controlled Trials, and Cochrane Central Database of Systematic Reviews databases, was performed for relevant articles published between January 2010 and January 2021. The following Medical Subjects Heading were used: “female/woman”, “urinary tract infection”, and “D-mannose”. Only clinical studies, systematic reviews, and meta-analyses reporting efficacy or safety data on D-mannose versus placebo or other competitors were selected for the present review. Studies published only as abstracts and reports from meetings were excluded. In addition, reference sections of the identified publications were also screened. Only publications in English were considered. Evidence was limited to human data, and, therefore, data from animal studies were excluded in the review. The title and abstract of each article were reviewed for their appropriateness and relevance. After removal of duplicates, two authors (CDN, VF) independently screened the titles and abstracts of 160 records for eligibility. The initial list of selected papers was further enriched by individuals’ suggestions from the coauthors of the present review. Reference lists of the selected articles/systematic reviews/metanalysis were also screened in order to identify other possible relevant studies using the same criteria of the initial search. Selected studies were organized in two categories according to the presence or not of a competitor to D-mannose. 

## 3. Evidence Synthesis

The search strategy allowed us to identify 160 clinical studies. After exclusion of non-pertinent studies/articles, 13 studies were analyzed (Table 1). In detail, six were randomized controlled trials (RCTs), one a randomized cross-over trial, five prospective cohort studies, and one a retrospective analysis. Seven studies compared D-mannose to placebo or other drugs/dietary supplements. Six studies evaluated the efficacy of D-mannose, comparing follow-up data with the baseline. 

### 3.1. D-Mannose: UTI Prevention and Treatment

In 2010, Efros et al. published a prospective, dose-escalation trial in women with a history of rUTIs. The authors tested an oral liquid dietary supplement composed of cranberry, ascorbic acid, D-mannose, fructo-oligosaccharides, and bromelain (UTI-STAT) [12]. The UTI-STAT was administered orally at 15, 30, 45, 60, and 75 mL daily for 12 weeks. The primary endpoints were safety, tolerability, and maximal tolerated dose. The secondary endpoints were efficacy with regard to rUTI and quality-of-life (QOL) symptoms. Only 23 patients were analyzed, and the maximal tolerated dose was 75 mL/d. Therefore, the recommended dose was set at 60 mL/d. Only 9% of the treated patients reported rUTIs, with a significant improvement of QoL regarding the physical functioning domain and role limitations from the physical health domain. In 2014, Vicariotto et al. assessed in a prospective, non-controlled study the effectiveness of an association of a cranberry dry extract, D-mannose, a gelling complex composed of the exopolysaccharides produced by *Streptococcus thermophilus* ST10 (DSM 25246), and tara gum, as well as 2 microorganisms *Lactobacillus plantarum* LP01 (LMG P-21021) and *Lactobacillus paracasei* LPC09 (DSM 24243) in women affected by acute uncomplicated cystitis in a pilot prospective study [13]. Thirty-three premenopausal, nonpregnant women diagnosed with acute uncomplicated cystitis took two doses per day during the first month and then continued with one sachet per day until the 60th day. The study showed a reduction of nitrites and leukocyte esterase during follow-up, and significant improvement of uncomfortable symptoms reported by 33 evaluated women with acute cystitis was suggested [13]. In 2016, Domenici et al. tested in 43 women the efficacy of D-mannose alone in the treatment of acute UTI and its possible role in the management of recurrences. D-mannose was administered twice daily for 3 days and then once a day for 10 days. In this prospective, non-comparative study, the authors observed, 15 days after D-mannose administration, a significant improvement of the majority of symptoms. Interestingly, one month after diagnosis, patients were consecutively randomized in two groups. In detail, 22 women received prophylaxis with D-mannose, and 21 remained untreated. The mean time to UTIs onset was 43 days (± 4.1 SD) in the group undergoing prophylaxis, and 28 (± 5.4 SD) in the other one (*p* = 0.0001). Treatment did not present any side effect also in a long-term schedule [14]. 

In 2017, Phe et al. performed an open-label feasibility, prospective study enrolling both men or women with multiple sclerosis and a history of rUTIs. The participants were treated with D-mannose powder 1.5 g twice daily for 16 weeks and were instructed to monitor suspected UTIs at home using urine dipsticks [15]. The 22 enrolled patients showed excellent compliance rates for using D-mannose and dipsticks for testing suspected UTIs, which were 100% and 90.2%, respectively. The use of oral D-mannose was associated with a significantly decrease of the number of monthly proven UTIs, in both catheter users and non-users. Notably, no adverse effects were reported. This study demonstrated the feasibility and the safety of oral D-mannose use in the setting of patients affected by multiple sclerosis [15]. In 2017, Marchiori et al. evaluated the effectiveness of the N-acetylcysteine, D-mannose, and a *Morinda citrifolia* fruit extract (NDM) associated with an antibiotic therapy to reduce the persistence of rUTI in 60 women, who were breast cancer survivors [16]. The authors analyzed retrospectively a group of 40 women treated with antibiotic therapy associated with NDM for six months and a second group of 20 patients treated with antibiotics alone. Patients treated with the association of antimicrobials and non-antimicrobials showed a significant reduction in bacteria-positive urine cultures and improvement in symptoms such as urinary urgency, frequency, urge incontinence, recurrent cystitis, bladder and urethral pain in comparison with the women treated only with antibiotics [16]. In 2018, Genovese et al. published a randomized three-arm parallel-group study enrolling 72 women with acute UTI and a history of recurrent cystitis. The three groups were treated with oral D-mannose, which was associated with berberine, arbutin, and birch in group A, with berberine, arbutin, birch, and forskolin in group B, and with proanthocyanidins in group C. The study duration was 12 weeks. The authors reported that patients in groups A and B had a lower incidence of episodes of recurrent cystitis during treatment and follow-up in comparison with those enrolled in group C [17]. In 2018, Del Popolo and Nelli explored the effectiveness of a combination of D-mannose, salicin, and *Lactobacillus acidophilus* (La-14) in patients complaining of recurrent symptomatic cystitis [18]. Notably, this study enrolled a consistent number of patients affected by neurogenic bladder due to spinal cord injury or multiple sclerosis. In total, 78 patients received an initial five-day regimen consisting of an oral combination of 1000 mg of D-mannose plus 200 mg of dry willow extract (salicin) three times daily (attack phase, morning, midday, evening, always on a full stomach), followed by seven days with 700 mg of D-mannnose plus 50 mg (1 × 109 CFU) of *L. acidophilus* (La-14) twice daily (maintenance treatment, morning and evening on a full stomach). This latest combination (D-mannose plus La-14) was repeated at the same dosage for 15 days each month for two months. Patients’ symptoms were evaluated through a three-day bladder diary and a Visual Analogic Scale (VAS). The authors reported significant symptoms improvement in both groups, including patients with or without neurogenic bladder. In particular, after treatment, the VAS scores decreased from 8.07 ± 1.70 to 4.74 ± 2.07 (*p* = 0.001) in non-neurological patients (group A) and from 7.21 ± 1.90 to 3.74 ± 3.12 (*p* = 0.001) in neurological patients (group B). A significant reduction of daily frequency was noted in both groups: from 14 ± 3 to 7 ± 3 (*p* = 0.001) in group A and from 15 ± 3 to 8 ± 3 (*p* = 0.001) in group B. A reduction of incontinence episodes in Group A patients was observed, as well as in 12/39 of Group B patients. Interestingly, no significant side effects were reported during the study period [18]. 

### 3.2. D-Mannose Versus Antibiotics or Polyphenols: UTI Prevention and Treatment

In 2014, Kranjcec et al. performed a randomized controlled trial to test the efficacy of D-mannose in rUTI prevention (Table 2.). After the initial antibiotic treatment of acute cystitis, 308 women with a history of rUTI were randomly allocated to three different groups. One group received a prophylaxis with 2 g of D-mannose powder in 200 mL of water daily for 6 months [19]. A second group received 50 mg of nitrofurantoin daily, and a third group did not receive prophylaxis. The number of rUTI was 14.6% in the D-mannose group, 20.4% in the nitrofurantoin group, and 60.8% in the no-prophylaxis group (*p* < 0001). Moreover, patients in the D-mannose group and nitrofurantoin group had a significantly lower risk of recurrent cystitis episodes during prophylactic therapy compared to patients in the no-prophylaxis group (RR 0.239–95% CI 0.146–0.392; *p* < 0.0001 and RR 0.335–95% CI 0.222–0.506; *p* < 0.0001). Notably, the difference between D-mannose and nitrofurantoin groups was not significant. Interestingly, D-mannose and nitrofurantoin prophylaxis remained independent variables in the prevention of rUTI also when the analysis was adjusted for age, history of UTI, and intercourse frequency. As expected, patients treated with D-mannose showed a significantly lower risk of side effects in comparison with those receiving prophylaxis with nitrofurantoin. In detail, D-mannose was responsible for diarrhea in 7.8% of cases. Nitrofurantoin was generally well tolerated but responsible for diarrhea in 9.7% of patients, nausea in 5.8%, headache in 2.9%, and vaginal burning in 8.7% [19]. Data reported in the previous RCT are very interesting and showed that D-mannose may be useful for UTI prevention. In 2014, Porru et al. evaluated the efficacy of D-mannose in the treatment and prophylaxis of rUTIs. The authors enrolled 60 women with acute symptomatic UTI and three or more with rUTI during the preceding 12 months. The patients were randomly assigned to two groups. The group treated with antibiotic received a five-day therapy with trimethoprim/sulfamethoxazole twice a day. The patients allocated to the D-mannose group received 1 g three times a day, every 8 h for 2 weeks and subsequently 1 g twice a day for 22 weeks. The mean time to rUTI was 52.7 days for the antibiotic group and 200 days for the group treated with oral D-mannose [20]. In 2017, Palleschi et al. compared the administration of an association of D-mannose, N-acetylcysteine (NAC) and *M. citrifolia* extract to antibiotic therapy in the prophylaxis of UTIs potentially associated with urodynamic examination [21]. The study included 42 men and 38 women who were randomized to receive non-antimicrobial prophylaxis versus antibiotic treatment with Prulifloxacine. Ten days after the urodynamic study, no differences between the two groups regarding the incidence of UTI were detected. The authors suggested the use of these nutraceutical agents is a good alternative in the prophylaxis of UTI associated with urological procedures [21]. In 2017, De Leo et al. tested the efficacy of a dietary supplement (Kistinox^®^ Forte sachets) containing cranberry, propolis extract, and D-mannose in 150 women suffering from rUTIs. In total, 100 patients were randomized to the treatment with one sachet per day during the first 10 days of the month for 3 months, and 50 were assigned to the control group and did not received any treatment [22]. The authors observed a statistically significant benefit in the treatment group, characterized by a reduction of dysuria and urination frequency. Moreover, this dietary supplement was well tolerated for the three-month period of the study [22]. In 2018, Salinas-Casado et al. compared the efficacy and safety of a dietary supplement consisting of 2 g of D-mannose (Manosar^®^) associated with proanthocyanidin (PAC) (140 mg), ursolic acid (7.98 mg), A, C, and D vitamins, and the oligoelement zinc, versus 240 mg of PAC alone in women with rUTI in a multicenter randomized double-blind study [23]. After a 24-week follow-up, the percentage of UTI was 24% in the D-mannose group and 45% in the PAC group (*p* < 0.05). The disease-free time for the Manosar^®^ group was 95 days, while this time was 79 days for the PAC group. The incidence of side effects was low. Diarrhea was the most frequent side effect in both groups. According to previous data, D-mannose (oral, once a day) appeared more effective than single-dose PAC (240 mg daily orally) in the prevention of recurrent UTI in women. In 2020, Salinas-Casado et al. published data of a randomized controlled trial comparing a food supplement containing a D-mannose -ike active ingredient (Manosar^®^) with another preparation in which the active ingredient was PAC [24]. These polyphenols, similar to D-mannose, help prevent the adhesion of certain harmful bacteria, including *E. coli*, associated with urinary tract infections onto cell walls. This multicenter, double-blind study was carried out for 24 weeks in a total of 184 randomized women with a history of rUTIs, without evidence of complication. In detail, 90 patients received Manosar^®^, and 94 isolated PAC. The percentage of patients developing UTI due to *E. coli* was 27.7% in the first group and 50% in the second one (*p* < 0.01). Notably, patients treated with Manosar^®^ showed a longer time free of rUTIs in comparison with those treated with isolated PAC. Therefore, data of this RCT support the role of D-mannose with respect to PAC in preventing rUTIs in women [24]

## 4. D-Mannose: Adverse Events

D-mannose is well tolerated, with few reported adverse events. Particularly, the most frequent adverse event is diarrhea, reported in about 8% of patients receiving 2 g of D--mannose for at least 6 months. The reported adverse event is mild, not leading to treatment discontinuation. In a comparative study, patients in a D-mannose group had a significantly lower risk of adverse events when compared to those in a nitrofurantoin group (RR 0.276, 95% CI 0.132–0.574, *p* = 0.0001) [19].

## 5. Discussion

Data of the present non-systematic review of the literature confirmed the potential role of oral D-mannose in the reduction of the risk of rUTIs in women. Notably, data available in the literature highlight the tolerability of D-mannose, with minimal side effects corresponding, mainly, to diarrhea. However, the available data on the effectiveness and safety of oral D-mannose alone or in combination with other substances are strongly limited by the poor quality of the available studies, regardless of the study design.

D-mannose binds to the tip of type 1 pili and saturates adhesin FimH, blocking bacterial adhesion to the urothelium and the signaling cascade responsible for the induction of uropathogenic urothelial invasion [5,25]. Moreover, recently, Zhang et al. suggested that D-mannose could play a role as an immune modulator [26]. Although type 1 fimbriae were extensively studied in *E. coli*, type 1 pili have been documented in several other uropathogens commonly involved in rUTIs [10,11]. 

Recently, Lenger et al. performed a systematic review of the literature and a meta-analysis of the available studies comparing D-mannose to placebo or antibiotic therapy, with the aim to demonstrate D-mannose role in the prevention of rUTIs. In particular, this meta-analysis showed a significant advantage of D-mannose versus placebo (RR 0.23–95%CI 0.14–0.37) and overlapping results in comparison with antibiotic treatment (RR 0.39–95%CI 0.12–1.25) [1]. This meta-analysis representing the high level of evidence currently available in the literature, also highlighted the limited side effects of D-mannose. However, only very few studies reported appropriately the side effects after assumption of D-mannose. Only three comparative studies were included in the previous meta-analysis [14,19,20]. 

In our non-systematic review of the literature, other three comparative studies were selected [21,23,24]. While Palleschi et al. evaluated a specific setting of male and women who underwent urodynamic examination, Salinas-Casado et al. performed two RCTS enrolling only women with rUTIs. The RCT published by Palleschi et al. showed that D-mannose can offer equivalent results to antibiotics in the prevention of UTIs in patients who underwent urodynamic examination, reinforcing the conclusions of the previous meta-analysis [1,21]. Interestingly, Salido-Casado et al. in 2018 demonstrated that the association of D-mannose with PAC worked significantly better in comparison with PAC alone, without differences in side effects in the two groups of patients examined [23]. These data are strongly supported by a more recent, multicenter RCT. Therefore, considering the last two RCTs, we can affirm that D-mannose should be preferred to PAC for a non-antimicrobial prophylaxis of women with rUTIs. This last information is additional in comparison with the recent meta-analysis by Lenger et al. [1]. 

Although our non-systematic review of the literature identified five RCTs evaluating the efficacy and safety of D-mannose against placebo, antibiotics, or PACs, the quality of the collected studies is very low, generating, consequently, a weak grade of recommendations. Therefore, we believe that further well-designed studies are urgently needed to definitively support the role of D-mannose in the management recurrent UTIs in women. Another open issue is if D-mannose works better in combination with others dietary supplements. Beyond the association with PACs, it could be interesting to further test its association with probiotics (*Lactobacillus*). The only study analyzing this association is strongly limited by the setting of patients in which the study was performed and by the absence of a control group [18]. Notwithstanding all the limitations of the current studies on the role of D-mannose in the prevention and treatment of rUTI, the evidence summarized in this review offers new insights for the non-anti-microbial prophylaxis of rUTI. Particularly, our findings support and confirm that D-mannose reduces the incidence of rUTIs and determines a longer interval between UTI episodes, with a significant improvement in patients‘ quality of life [4], and therefore, its administration could be considered as one of the possible strategies to be used or to be investigated for the prevention of recurrent UTI in women (Table 3.) 

## 6. Conclusions

D-mannose is effective and safe in the non-antimicrobial prophylaxis of rUTIs in women. However, the evidence comes from poor-quality studies, and further well-designed studies evaluating D-mannose alone or in association with other dietary supplements versus placebo or antibiotics must be urgently planned before D-mannose implementation in the clinical practice. In fact, despite D-mannose being highly prescribed in real-life practice, alone or in combination with several dietary supplements, for the prevention or management of rUTI, the European Urological Guidelines on urinary tract infections do not recommend its routinely use outside clinical trials. The possible role of D-mannose in *E. coli* metabolism has been recently investigated. D-mannose has no impact on bacteria metabolism, either on antibiotic activity or bacteria viability. Furthermore, D-mannose does not present an antibiotic-like activity, considering that it does not induce FimH variants that can modify bacterial adhesiveness [9]. In the recent years, some authors have evaluated the use of synthetic D-mannose analogs [9,27,28,29]. Their ability to bind to *E. coli* may improve the management of patients with recurrent UTIs. The development of anti-adhesive molecules represents nowadays a promising area of research in UTI prophylaxis. However, their use in clinical practice is still to be defined and should be investigated in further clinical trials.

## Figures and Tables

**Table 1 antibiotics-10-00373-t001:** Summary of non-comparative studies evaluating D-mannose administration.

Author, Year	Study Design	Study Population	Inclusion Criteria	Treatment	Main Results
Efros et al., 2010	Prospective	23 Women 18–75 years	Recurrent UTIs	-Proantinox °	9% UTI Improvement in QoL
Vicariotto, 2014	Prospective	35 Women Premenopausal	Acute symptomatic UTI	D-mannose 250 mg Tara Gum 250 mg Proanthocyanidin cranberry extract + Probiotics *	Reduction in Nitrites and Leucocytes
Domenici, 2016	Prospective	43 Women	Acute symptomatic UTI	D-mannose (1.5 g), sodium bicarbonate, sorbitol, and silicon dioxide.	Reduced number of UTIs.
Phe, 2017	Prospective	22 Men and women with multiple sclerosis	Recurrent UTIs	D-mannose powder (1.5 g)	Decrease in the number of UTIs.
Marchiori, 2017	Retrospective	60 breast cancer women	Breast cancer survivors	Antibiotics + Mannose vs. Antibiotics	Greater efficacy of combination treatment in preventing UTIs
Genovese, 2018	Randomized	72 Women	Recurrent UTIs	A: D-mannose Berberine, Arbutin and Birch B: D-mannose, Berberine, Arbutin, Birch, Forskolin C: D-mannose, Proanthocyanidis	A and B > C in terms of number of UTI
Del Popolo, 2018	Prospective	78 Patients Neurogenic bladder	Recurrent UTIs	D-mannose (1 g) + dry willow extract (200 mg)	D-mannose improved VAS scores and frequency of incontinence episodes.

° Proantinox: cranberry concentrate [4:1], ascorbic acid, D-mannose, fructo-oligosaccharides, and bromelain) per 30 mL (UTI-STAT with Proantinox, Medical Nutrition USA, Englewood, NJ); * Probiotics: 2.5 billion live cells of *Lactobacillus plantarum* LP01 (LMG P-21021); 1 billion viable cells each of *Lactobacillus paracasei* LPC09 (DSM 24243), and *Streptococcus thermophilus* ST10 (DSM 25246); UTIs: urinary tract infections, QoL: quality of life.

**Table 2 antibiotics-10-00373-t002:** Summary of comparative studies evaluating D-mannose administration.

Author, Year	Study Design	Study Population	Inclusion Criteria	Treatment	Main Results
Kranjcec et al., 2014	Randomized	308 Women 18–75 years	Recurrent UTIs	- D-mannose - Nitrofurantoin - None	D-mannose (15%) and Nitrofurantoin (20%) groups presented a lower risk of UTIs when compared to controls (61%).
Porru, 2014	Randomized	60 Women	Recurrent UTIs	- Trimetoprim/Sulfamethoxazole - D-mannose	Mean time to UTI 52 vs. 200 days favors D-mannose
Palleschi, 2017	Randomized	42 Men and 38 Women	Post Urodynamic	- D-mannose, N-acetylcisteintand Morinda citriofula - Antibiotic	No difference in terms of UTis post- urodynamics.
De Leo 2017	Randomized	150 Women	Recurrent UTIs	- Kistinox (Carnberry, Propolis, D-Mannose) - No treatment	Reduction of dysuria and frequency.
Salinas-Casado 2018	Randomized	150 Women	Recurrent UTIs	- D-mannose 2 g + PAC 140 mg - PAC 240 mg	95 days vs. 79 days disease-free favor the D-mannose group.
Salinas-Casado 2020	Randomized	184 Women	Recurrent UTIs	- D-mannose 2 g - PAC 240 mg	27% vs. 50% of UTIs favors the D-mannose group.

PAC: proanthocyanidins.

**Table 3 antibiotics-10-00373-t003:** Overview of potential treatments beyond antibiotics for recurrent cystitis in women.

Characteristics	Hormonal Replacement	Immunoactive Prophylaxis	*Lactobacillus*	Cranberry	Hyaluronic Acid	D-mannose
Data maturity	Solid	Solid	Poor	Poor	Poor	Poor
Route of Administration	Vaginal	Oral	Oral	Oral	Endovesical Oral	Oral
Adverse Events	Vaginal discomfort, irritation, burning, itching	None	None	None	Burning during instillation (78%)	Diarrhea (<8%)
EAU Guidelines on Urinary Tract Infections	Recommended for post-menopausal women (weak)	More effective than placebo (strong)	Further trials are needed before recommendation for or against its use	Contradictory results. No recommendation	No recommendation	Data indicative but not sufficient for recommendation

## Data Availability

Data are available upon request of the readers.

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
