# Peer review of "Role of D-Mannose in the Prevention of Recurrent Uncomplicated Cystitis: State of the Art and Future Perspectives"

_antibiotics, 2021, doi:10.3390/antibiotics10040373_

Round 1

Reviewer 1 Report

The authors performed a non-systematic literature analysis about the efficacy of D-mannose for the treatment of (recurrent) urinary tract infections. They found that there is evidence that D-mannose can prevent UTIs.

This is an interesting study and the evidence is well-compiled. It is intriguing that while used in practice often, D-mannose is not recommended by European guidelines.

One comment to make would be why the authors chose a non-systematic approach which is prone to omit studies that may influence the results.

Author Response

The authors performed a non-systematic literature analysis about the efficacy of D-mannose for the treatment of (recurrent) urinary tract infections. They found that there is evidence that D-mannose can prevent UTIs.

This is an interesting study and the evidence is well-compiled. It is intriguing that while used in practice often, D-mannose is not recommended by European guidelines.

We thank the reviewer for the positive comments.

One comment to make would be why the authors chose a non-systematic approach which is prone to omit studies that may influence the results.

We thank the reviewer for his/her comment and for the possibility to better explain these aspects. We agree with the reviewer that systematic reviews represent a higher level of evidence however it is no always possible to perform such kind of studies due to the lack of resources or evidence in the literature. Our objective was to build a scoping review. The objective of our scoping review was to identify the types of available evidence on D- mannose, the key characteristics of the evidence and to analyze knowledge gaps on the subject. Furthermore, our objective was not to answer a particular question rather to provide an overview or map of the evidence. Our review should serve as a precursor for a systematic review on the subject. (https://doi.org/10.1186/s12874-018-0611-x).

See Introduction section:

‘Moreover, the objective of our scoping review was to identify the types of available evidence on D- mannose, the key characteristics of the evidence and to analyze knowledge gaps on the subject in order to provide an overview of the evidence. Furthermore, our review could be considered a precursor of a future systematic review.

Reviewer 2 Report

The authors present an exhaustive and well organized description of their limited literature analysis of the use of D-mannose and other dietary components for the non-antibiotic treatment of rUTIs. The work constitutes a valuable analysis of the clinical outcome on the use of D-Mannose and would be useful to clinicians. The manuscript is well written by in our opinion lacks citation of literature that, unfortunately fall outside their search. For instance, they have not discussed the used of modern medicinal chemistry in the use of D-mannose analogs having strong binding affinities against the E. coli FimH adhesion, which would reinforce their analysis (on top of ref 25, see below). In this regards, some additional very recent references should be added, even just in the concluding remarks, that would widen the scope of forthcoming reviews on this important disease situation. In addition, several papers have reported that D-mannose constituted an alternative nutrient for the growth of the E. coli (and likely other pathogens) in D-glucose deprived culture. Even though these additional data may be argued to fall outside of the scope of the manuscript, they would definitely improve the scientific coverage of this review. Recommended citations : DOI: 10.3390/antibiotics9070397; Chembiochem 21, 1–18 ; DOI: 10.3390/molecules25020316; DOI: 0.1021/acs.accounts.8b00397, https://doi.org/10.1016/j.drudis.2021.02.025; doi: 10.15761/FDCCR.1000115;

Author Response

Reviewer 2

The authors present an exhaustive and well organized description of their limited literature analysis of the use of D-mannose and other dietary components for the non-antibiotic treatment of rUTIs. The work constitutes a valuable analysis of the clinical outcome on the use of D-Mannose and would be useful to clinicians.

We thank the reviewer for the positive comments.

The manuscript is well written by in our opinion lacks citation of literature that, unfortunately fall outside their search. For instance, they have not discussed the used of modern medicinal chemistry in the use of D-mannose analogs having strong binding affinities against the E. coli FimH adhesion, which would reinforce their analysis (on top of ref 25, see below). In this regards, some additional very recent references should be added, even just in the concluding remarks, that would widen the scope of forthcoming reviews on this important disease situation. In addition, several papers have reported that D-mannose constituted an alternative nutrient for the growth of the E. coli (and likely other pathogens) in D-glucose deprived culture. Even though these additional data may be argued to fall outside of the scope of the manuscript, they would definitely improve the scientific coverage of this review.

Recommended citations: DOI: 10.3390/antibiotics9070397; Chembiochem 21, 1–18; DOI: 10.3390/molecules25020316; DOI: 0.1021/acs.accounts.8b00397, https://doi.org/10.1016/j.drudis.2021.02.025; doi: 10.15761/FDCCR.1000115;

We thank the reviewer for the comments and for the possibility to improve our manuscript. According to the reviewers’ comment a paragraph has been added to improve our manuscript and the suggested references have been added.

See Conclusion section:

The possible role of D-mannose in E-Coli metabolism has been recently investigated. D-mannose has no impact on bacteria metabolism neither on antibiotic activity or bacteria viability. Furthermore,  D-mannose does not present an antibiotic like activity considering that it does not induce FimH variants that can modify bacterial adhesiveness[9].  In the recent years some authors have evaluated as well the use of D-mannose analogs created by the modern chemistry[9,27–29]. Their ability to bind to E-Coli may improve the management of patients with recurrent UTIs. The development of anti-adhesive molecules represents nowadays a promising area of research in the UTI’s prophylaxis. However, their use in clinical practice is still to be defined and should be investigated in further clinical trials.